# ECO: Energy-Constrained Operator Learning for Chaotic Dynamics with Boundedness Guarantees

## Abstract

Chaos is a fundamental feature of many complex dynamical systems, including weather systems and fluid turbulence. These systems are inherently difficult to predict due to their extreme sensitivity to initial conditions. Many chaotic systems are dissipative and ergodic, motivating data-driven models that aim to learn invariant statistical properties over long time horizons. While recent models have shown empirical success in preserving invariant statistics, they are prone to generating unbounded predictions, which prevent meaningful statistics evaluation. To overcome this, we introduce the **Energy-Constrained Operator (ECO)** that simultaneously learns the system dynamics while enforcing boundedness in predictions. We leverage concepts from control theory to develop algebraic conditions based on a learnable energy function, ensuring the learned dynamics is dissipative. ECO enforces these algebraic conditions through an efficient closed-form quadratic projection layer, which provides provable trajectory boundedness. To our knowledge, this is the first work establishing such formal guarantees for data-driven chaotic dynamics models. Additionally, the learned invariant level set provides an outer estimate for the strange attractor, a complex structure that is computationally intractable to characterize. We demonstrate empirical success in ECO's ability to generate stable long-horizon forecasts, capturing invariant statistics on systems governed by chaotic PDEs, including the Kuramoto–Sivashinsky and the Navier–Stokes equations.

## 1 Introduction

**Chaotic dynamical systems.** Chaos arises in a wide range of physical systems, including weather models (Lorenz, 1963) and fluid dynamics (Kuramoto, 1978; Sivashinsky, 1988). A hallmark of chaotic systems is their sensitivity to initial conditions. That is, small perturbations can cause exponential divergence between trajectories, making precise long-term prediction intractable. Despite this unpredictability, many chaotic physical systems are dissipative, meaning their trajectories converge to a lower-dimensional invariant set, often referred to as the strange attractor (Stuart & Humphries, 1998). Once in this attractor, the system exhibits ergodicity, where the trajectory will eventually visit every state on the attractor (Guckenheimer & Holmes, 2013). The ergodic behavior of dissipative chaotic systems, coupled with the intractability in predicting exact pointwise trajectories, makes capturing the statistical behavior of a system on the attractor a natural goal.

Recent data-driven methods have achieved impressive empirical success in constructing models that both speed up inference and capture the long-term invariant statistics of dissipative chaotic systems. These approaches vary widely in terms of structural assumptions and model complexity. At one end of the spectrum, structured nonlinear regression techniques employ physically motivated multi-level models to efficiently fit time series data (Majda et al., 2001; Majda & Harlim, 2012; Goyal et al., 2023). At the other end, deep learning methods use the expressive capabilities of neural networks to directly learn complex chaotic dynamics from raw data, often embedding physical system knowledge into the architecture or regularization strategies (Li et al., 2020; Tang et al., 2024; Raissi et al., 2019; Brunton & Kutz, 2022; Lu et al., 2021; Kochkov et al., 2021; Page et al., 2024). Hybrid models bridge these approaches by using autoencoder architectures to project data into latent spaces where the dynamics become simpler—drawing inspiration from Koopman theory (Koopman, 1931), Dynamic mode decomposition (Kutz et al., 2016), PCA (Pearson, 1901), etc. Recurrent sequential

models have been developed to enhance stability and accuracy by incorporating more input history beyond a single time step (Mikhaeil et al., 2022; Vlachas et al., 2018; Sangiorgio & Dercole, 2020). Among these, a specific recurrent architecture designed for time series prediction, called reservoir computing, has shown strong performance in capturing invariant statistics and reconstructing attractors (Lu et al., 2018; Vlachas et al., 2020; Bollt, 2021).

Data-driven methods for modeling chaotic systems often adopt an autoregressive framework, where a model predicts the next time step based on the current state to generate long-term trajectories and capture statistical properties. However, this approach can suffer from accumulated errors, causing trajectories to drift away from the training distribution. In chaotic settings, such drift can result in unbounded or nonphysical predictions, ultimately corrupting statistical estimates derived from the generated trajectories. Structured models, such as multi-level quadratic regression, have been theoretically shown to exhibit pathological instability in their statistical behavior (Majda & Yuan, 2012). In the context of recurrent neural networks (RNNs), Mikhaeil et al. (2022) demonstrated that training such models on chaotic systems leads to gradient divergence, highlighting a fundamental limitation. While the theoretical understanding of machine learning models remains limited, empirical evidence shows that these models often produce divergent trajectories, ultimately resulting in unreliable statistical predictions. This issue has also been observed in advanced time-series modeling techniques, including Reservoir Computing and Fourier Neural Operators (Lu et al., 2018; Pathak et al., 2017; Li et al., 2022). As such, a central challenge across both physics-informed and purely data-driven approaches is the difficulty of maintaining stable long-term forecasts.

**Learning with hard constraints.** Recent efforts in machine learning have explored hard-constrained neural networks as a means to enforce physical or structural constraints on model predictions. These methods can guarantee satisfaction of linear equality constraints (Chen et al., 2024; Balestriero & LeCun, 2023) or combinations of linear equality and inequality constraints (Min et al., 2024; Flores et al., 2025). Often, these approaches employ either a specific parameterization of a given network or a network-agnostic, closed-form projection layer that maps neural network outputs to a feasible set. Such constraints have been successfully applied to enforce physical constraints such as conservation laws (Chen et al., 2024) as well as stability objectives in learned dynamics (Min et al., 2023). However, these methods are limited to affine constraints. (Tordesillas et al., 2023) introduce a network parameterization method to enforce certain convex constraints, but is limited to input-independent constraints. Other methods (Lastrucci & Schweidtmann, 2025) use iterative Newton-like approaches to push model outputs towards satisfaction of nonlinear equality constraints but can be expensive and are not guaranteed to converge under certain conditions. There remains a gap in the existing literature for closed-form guaranteed satisfaction of nonlinear constraints, which are of particular interest for physical systems. In the context of chaotic systems, enforcing boundedness in predictions can be useful for avoiding trajectory blow-up.

**Our contributions.** We introduce the Energy-Constrained Operator (ECO), a framework for learning chaotic dynamics that guarantees long-term stability by construction. By integrating control-theoretic principles, ECO simultaneously learns the dynamics and a stabilizing energy function. To our knowledge, this is the first work to provide explicit formal guarantees on the boundedness of predicted trajectories for such systems. Our key contributions include[1]:

- **A Learnable Energy-Based Constraint:** We derive computationally efficient, algebraic dissipative conditions based on a learnable quadratic Lyapunov function. This allows the model to discover a stabilizing invariant set from data without requiring prior knowledge of the system, which is an outer-estimate for the attractor.

- **A Convex Quadratic Projection Layer:** We design a projection layer that enforces a general convex quadratic constraint, including the dissipativity condition. The layer is network-agnostic, computationally efficient, and fully differentiable.

- **Theoretical Boundedness Guarantees:** We provide a formal proof that our learned model is globally asymptotically stable, ensuring all prediction trajectories remain bounded and converge to the learned invariant set.

- **Empirical Results:** We validate ECO on challenging chaotic benchmarks, including the Kuramoto–Sivashinsky and Navier–Stokes equations. Our model produces stable long-horizon forecasts that accurately reconstruct statistical properties of the strange attractors.

---

[1]For reproducibility, our source code is available at `https://anonymous.4open.science/r/eco_pde-2655/`

## 2 PROBLEM FORMULATION

Consider a chaotic dynamical system described with a PDE of the form,

$$\partial_t w = F\left(x, w, \partial_x w, \partial_{xx} w, ...\right), \qquad (t, x) \in [0, T] \times \mathbb{X}$$
$$w(0, x) = w^0(x), \qquad x \in \mathbb{X} \tag{1}$$
$$B\left[w\right](t, x) = 0, \qquad (t, x) \in [0, T] \times \partial\mathbb{X}$$

Here, $w(t, x)$ represents the $n$-dimensional state of the dynamical system at any time $t \in [0, T]$ and position $x \in \mathbb{X} \subseteq \mathbb{R}^d$, $w_0(x)$ is the initial condition defined on the full spatial domain $x \in \mathbb{X}$, and $B\left[w\right](t, x)$ is the boundary condition defined on the spatial boundary $\partial\mathbb{X}$. We adopt a discrete-time formulation of this problem, and consider PDEs with solutions in $L^2$ space, i.e., $\mathbb{X} \subset L^2$,

$$w_{t+1}(x) = G(w_t(x), x), \qquad (t, x) \in \{0, 1, 2, ...N\} \times \mathbb{X} \tag{2}$$

where $w_t = w(t, \cdot) : \mathbb{X} \to \mathbb{R}$ represents the state of the dynamical system at any position $x \in \mathbb{X}$ at time step $t$. We focus on PDEs that govern dissipative chaotic systems, as many physically relevant chaotic systems inherently exhibit dissipative behavior.

The primary goal is to learn a neural operator $G^*(\theta)$ that emulates the true dynamics $G$. For the chaotic systems we study, long-term pointwise prediction is intractable due to sensitive dependence on initial conditions. While individual trajectories are unpredictable, these dissipative chaotic system's trajectories converge to a statistically invariant strange attractor. Consequently, a more feasible and meaningful objective is to learn an operator that preserves the statistical properties of the true dynamics over long horizons.

A significant challenge to achieving this objective is model stability. When an approximate operator $G^*(\theta)$ is applied iteratively to generate long-horizon trajectory forecasts, small prediction errors can be amplified exponentially, which accumulates and eventually causes predicted trajectories to diverge to unbounded values. This failure mode, also known as "finite-time blowup", prevents the model from capturing any meaningful long-term behavior Li et al. (2020); Lu et al. (2018).

To address this issue, our solution is to construct a model architecture that is dissipative by design. Since dissipativity is a fundamental property of the physical system, incorporating it as an inductive bias ensures trajectory boundedness without sacrificing expressivity. To achieve this rigorously, we first formalize the concept:

**Definition 1.** *We say that the system in (2) is **dissipative** if there exists a bounded (with respect to $L^2$ norm) and positively invariant set $M \subset L^2$ such that*

$$\lim_{t \to \infty} dist(w_t, M) = 0, \quad dist(w_t, M) = \inf_{y \in M} \|w_t - y\|$$

*In other words, every trajectory of the system will converge to $M$ asymptotically, and stays within $M$ once it enters. $M$ is said to be **globally asymptotically stable**.*

Intuitively, a dissipative system loses energy until its trajectories enter and remain within a bounded region $M$, which for chaotic PDEs is their strange attractor. However, Definition 1 is *descriptive* in the desired property of $M$, without providing a practical mechanism to verify or enforce it. Furthermore, the strange attractor $M$ itself is highly system-dependent and has been known computationally intractable to characterize (Stuart & Humphries, 1998; Milnor, 1985).

## 3 DISSIPATIVE DYNAMICS: A CONTROL-THEORETIC PERSPECTIVE

Our objective is to design a neural operator architecture that enforces dissipativity by construction. The first challenge is that Definition 1 is non-constructive, as it relies on the strange attractor $M$, which is computationally intractable to characterize. Therefore, we must establish an alternative, computationally efficient condition that can be directly enforced during training and inference.

For this purpose, we turn to a control-theoretic perspective and the concept of Lyapunov functions. These "energy-like" functions are extensively used to establish asymptotic stability of dynamical systems (Khalil, 2002) and naturally connect with the behavior of dissipative systems. Instead of

analyzing the complex attractor $M$, we can use a level set of a Lyapunov function as a tractable proxy for the bounded region.

We adapt this strategy in the following proposition, which generalizes the concept of asymptotic stability from a single equilibrium point to an entire level set, providing the practical conditions needed for our model design.

**Proposition 1** (set asymptotic stability). *For an infinite-dimensional dynamical system in (2), suppose there exists a non-negative-valued continuously differentiable function $V : L^2 \to \mathbb{R}_+$ and a constant $c > 0$, such that*

$$(i) \quad \forall w_t \notin M(c) = \{w \in L^2 : V(w) \le c\}, V(w_{t+1}) \le \alpha V(w_t), \quad 0 < \alpha < 1$$

$$(ii) \quad \forall w_t \in M(c) = \{w \in L^2 : V(w) \le c\}, V(w_{t+1}) \le c$$

$$(iii) \quad V \text{ is radially unbounded, i.e., } V(w) \to \infty \text{ as } \|w\| \to \infty$$

*Then the system (2) is dissipative, where the level set $M(c)$ is globally asymptotically stable.*

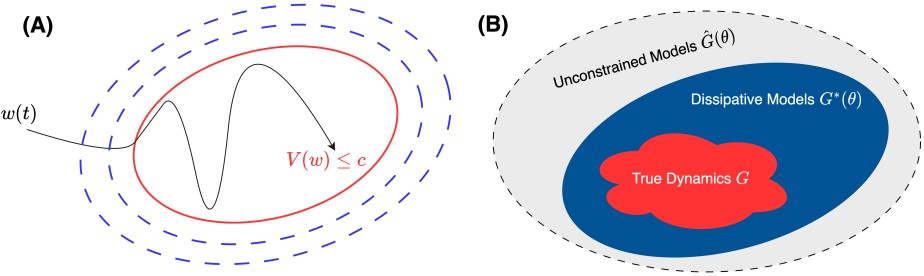

Figure 1: (A) An illustration of the conditions in Proposition 1, where the trajectory loses energy over time and enters an invariant level set. (B) Illustration of the fact that an inherently dissipative model would have an effectively smaller search space for parameters.

As illustrated in Figure 1(A), the conditions in Proposition 1 guide any solution starting outside the level set $M(c)$ to lose energy exponentially due to the $\alpha$ factor, entering the level set in finite time and remaining inside thereafter. A detailed proof for Proposition 1 is included in Appendix A.

Despite the simplicity of the algebraic conditions derived in Proposition 1, overall the conditions still obtain a form of "if-else" condition, which might not be straightforward to enforce in a neural network that requires differentiability for backpropagation. To resolve this issue, we unify conditions (i) and (ii) in the above proposition into the following single inequality constraint, which involves a ReLU activation and the $\alpha \in (0, 1)$ used in condition (3):

$$V(w_{t+1}) - \alpha \left[ V(w_t) + \text{ReLU} \left( c - V(w_t) \right) \right] \le 0 \tag{3}$$

Note that the reformulation here is equivalent, up to scaling the level set constant by $\alpha$. More specifically, the inequality (3) reduces to $V(w_{t+1}) \le \alpha V(w_t)$ when $V(w_t) \notin M(c)$, and reduces to $V(w_{t+1}) \le \alpha c$ when $V(w_t) \in M(c)$. As a result, under the assumption for $V$ made in Proposition 1, the reformulated constraint ensures that the system is dissipative and the level set $M(c)$ is globally asymptotically stable.

## 4 METHODOLOGY

To achieve the goal of building a neural network prediction model that ensures the trajectory it generates always stays bounded, we now introduce a framework that learns dissipative dynamics by design, based on the control-theoretic conditions derived in Section 3. As illustrated in Figure 1(B), learning an inherently dissipative prediction model conceptually limits the parameter search to a smaller space that is always aligned with physical properties of the true dynamics. Compared to unconstrained models, which might search over parameters that lead to unstable behaviors, our approach makes the training process more efficient.

Our methodology is built on two key components: 1) a learnable Lyapunov functional $V(w)$ that represents the system's energy, and 2) a custom dissipative projection layer that strictly enforces the constraint in (3). In addition to the learned model being dissipative, our framework is also able to produce an outer-estimate (the level set $M(c)$) for the complex strange attractor without any prior knowledge of the system's invariant statistics, which is known to be difficult to characterize. In what follows, we discuss the details of our architecture design and training procedure.

### 4.1 Architecture Design with Boundedness Guarantees

We propose a neural network architecture that simultaneously learns the dynamics operator in Equation (2) and an energy-like Lyapunov functional $V = \int_{\mathcal{D}} \left( Q \circ (w - w_c)^2 \right)(x)dx$, which together guarantee the dissipativity conditions in Proposition 1 through the construction of a dissipative projection layer. Following common practices in learning operators in function spaces (Lu et al., 2021; Kovachki et al., 2023; Li et al., 2022), we consider a discretized spatial domain where the queried spatial location $x \in \mathbb{X}$ is sampled from a finite set $\mathbb{X}_d$ consisting of $n$ grid points, i.e., $x \in \mathbb{X}_d \subset \mathbb{X}$ and the cardinality of $\mathbb{X}_d$ is $n$. As an example, if the spatial domain $\mathbb{X} = [0, 2\pi]$, a fixed grid on $\mathbb{X}$ can be $n$ evenly sampled points, $\mathbb{X}_d = \{k\frac{2\pi}{n-1} : k = 0, 1, ..., n - 1\}$. Under the grid setting, the function $w_t \in L^2$ can be effectively represented as an $n$–dimensional vector, which is a collection of solution values at every grid point $w_t := \{w(t, x) : x \in \mathbb{X}_d\} \in \mathbb{R}^n$. Consequently, the $L^2$ norm of $w_t$ is reduced to a standard 2–norm in $\mathbb{R}^n$, and the Lyapunov functional is reduced to $V(w) = (w - w_c)^T Q(w - w_c)$ where $Q \in \mathbb{S}_{++}$.

As illustrated in Figure 2(A), our model is composed of two learnable components:

1. An unconstrained dynamics emulator $\hat{G}$, which approximates the true dynamics operator $G$. The backbone model for the emulator $\hat{G}$ can be any neural operator that maps between function spaces. Here we choose to use DeepONet proposed in (Lu et al., 2021).

2. A quadratic Lyapunov functional $V(w) = (w - w_c)^T Q(w - w_c)$, which serves as the energy function. The learnable parameters include a positive definite matrix $Q$ of size $n$–by–$n$ and a center vector $w_c \in \mathbb{R}^n$.

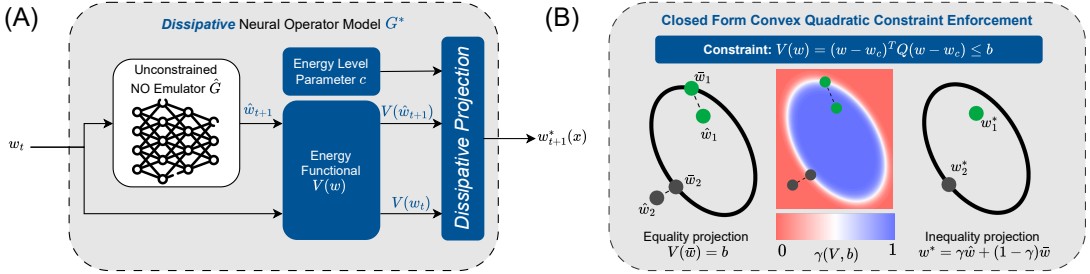

Figure 2: (A) An overview of the proposed model architecture. The input, current time solution $w_t$, is fed into an unconstrained neural operator (NO) emulator $\hat{G}$ to produce a preliminary prediction $\hat{w}_{t+1}$ and a learned energy functional $V$ to compute its energy $V(w_t)$. The dissipative projection layer modifies $\hat{w}_{t+1}$ to produce a final output $w^*_{t+1}$ that satisfies the dissipative energy constraint in (3). (B) Illustration of the convex quadratic projection for a constraint in the form of $V(w) \leq b$. The equality projection maps any point not on the ellipsoid boundary $w$ to a boundary point $\bar{w}$ in closed form (for both $w_1, w_2$). The quadratic projection is only active for when the constraint is violated, so $w_2$ is projected while $w_1$ is left unchanged.

These components are integrated into a *dissipative projection* layer, which modifies the output of the unconstrained emulator $\hat{G}$ to produce an operator $G^*$ that maps the current solution $w_t$ to the predicted solution at the next time step $w^*_{t+1}$. By construction, the *dissipative projection* layer ensures the condition in (3) is satisfied, which guarantees the predicted dynamical system $w^*_{t+1} = G^*(w_t)$ is dissipative. As a direct consequence of Definition 1, all trajectories generated by the operator $G^*$ in an autoregressive manner are guaranteed to be bounded. For a high dimensional output space, we elect to use a diagonal $Q$ matrix such that the projection can be computed efficiently.

## 4.2 CONVEX QUADRATIC PROJECTION LAYER

The quadratic form of the Lyapunov functional $V(w)$ motivates the development of a convex quadratic projection layer that projects the model predictions onto a feasible set of trajectories where the dynamics are dissipative. We introduce a differentiable convex quadratic projection layer shown in Figure 2(B) that can handle constraints of the form $(w - w_c)^T Q(w - w_c) \leq b$, where $b$ can be a constant or an arbitrary function of the model input. This form is equivalent to (3), with $b = \alpha \left[ V(w_t) + \text{ReLU}(c - V(w_t)) \right]$.

The convex quadratic constraint projection is illustrated in Figure 2. The general strategy is to define a projection $\bar{w}$ of the model output $\hat{w}$ onto the equality constraint $V(\bar{w}) = b$ and selectively project points that violate the constraint $V(\hat{w}) \leq b$ onto their respective equality projection. For positive definite $Q$, there exists an explicit form for the projection of $\hat{w}$ onto the equality constraint $V(\bar{w}) = b$. That is, for the quadratic Lyapunov function described in Section 4.1, the projection $\bar{w} = w_c + \sqrt{b} \left( L^T \right)^{-1} \frac{\hat{w} - w_c}{||\hat{w} - w_c||_2}$ satisfies the equation $V(\bar{w}) = b$, where $L$ is the Cholesky decomposition of $Q$ such that $LL^T = Q$. To ensure that this equality projection is only active when the constraint is violated, the final output $w^*$ is calculated as an interpolation between the projected ($\bar{w}$) and non-projected ($\hat{w}$) outputs.

$$w^*(x) = \gamma(b, V(\hat{w}))\hat{w}(x) + \left[1 - \gamma(b, V(\hat{w}))\right] \bar{w}(x) \tag{4}$$

Ideally, $\gamma(b, V(\hat{w}))$ is an indicator function that is 1 when $V(\hat{w}_t) \leq b$ and 0 when $V(\hat{w}_t) > b$. However, using an indicator function leads to non-differentiability, which prevents the model from learning a good energy functional $V$ effectively. Instead, we replace the indicator function with sigmoid as a smooth alternative $\gamma(b, V(\hat{w})) = \text{sigmoid} \left[ k(b - V(\hat{w})) \right]$.

## 4.3 THEORETICAL GUARANTEES

Our framework is designed to provide formal guarantees of stability and boundedness by construction. These guarantees stem from the convex quadratic projection layer. While this layer utilizes a sigmoid function as a continuous and differentiable relaxation of a strict indicator function, it maintains rigorous theoretical properties. The core idea is that this principled relaxation enforces the constraint on a controllably enlarged invariant set, whose size is governed by the sigmoid's steepness parameter. This is formalized in the following lemma.

**Lemma 1.** *For a quadratic positive definite Lyapunov function, the sigmoid relaxation of the projection in (4) maintains the boundedness of the projection output $w^*(x)$, with its energy upper bounded by $V(w^*(x)) \leq (1 + \delta)^2 b$, where $\delta = (2kb + 2\sqrt{2kb})^{-1}$ and $k$ is the sigmoid function parameter.*

The factor $\delta$ provides an explicit, non-asymptotic bound on the relaxation's cost as in the enlargement of the projected ellipsoid for any finite sigmoid function steepness $k$. This quantifiable enlargement allows us to enforce the conditions stated in Proposition 1 by choosing a contractivity factor $\alpha$ only marginally smaller than 1, preserving the model's expressiveness. This guarantee is formalized in the following theorem.

**Theorem 1.** *Let the learned dynamics be defined by the operator $w_{t+1}^* = G^*(w_t^*)$, which is composed of an unconstrained neural operator emulator $\hat{G}$ and a dissipative projection layer. Let the learned energy-like function be a quadratic Lyapunov functional $V(w) = (w - w_c)^T Q(w - w_c)$ with learnable center $w_c \in \mathbb{R}^n$ and a symmetric positive definite matrix $Q \in \mathbb{S}_{++}^n$. For a choice of $c > 1/\alpha$ and $0 < \alpha < [1 + (2k + 2\sqrt{2k})^{-1}]^{-2}$, the learned dynamical system is dissipative by construction, i.e., the level set $M(c)$ is globally asymptotically stable, and all trajectories generated by the learned dynamics are guaranteed to be bounded.*

**Practical Implications.** The condition in Theorem 1 is non-restrictive and is satisfied using fixed hyperparameters, obviating the need for extensive tuning. For all experiments, we fix the level-set scaling $c \gg 1$ and set $k = 100$, which yields the requirement $\alpha < 0.9913$. Our chosen value of $\alpha = 0.99$ comfortably satisfies this bound. Thus, our framework pairs a rigorous stability guarantee with the flexibility to learn nearly energy-preserving dynamics with a sufficiently large $k$ ($\alpha \to 1$ as $k \to \infty$), which is crucial for high-fidelity physical simulations.

Due to space constraints, the proofs for Lemma 1 and Theorem 1 are provided in Appendix A.

### 4.4 Training with Invariant Set Volume Regularization

We construct the training dataset as a collection of $N$ input-output pairs, denoted as $\{(w_i, w_{\text{next},i})\}_{i=1}^N$, where the input $w_i$ is the current time step solution and the output $w_{\text{next},i}$ is the next time step solution based on the true dynamics. During training, each input $w_i$ is mapped to a predicted output $w^*_{\text{next},i}$, and the loss is computed relative to the true next state $w_{\text{next},i}$. The dynamic loss is defined as the average mean squared error (MSE) between the predicted outputs $\{w^*_{\text{next},i}\}_{i=1}^N$ and the ground-truth future states $\{w_{\text{next},i}\}_{i=1}^N$.

While the convex quadratic projection layer enforces dissipativity and convergence to a level set $M(c)$, it does not inform how to choose an appropriate level set that characterizes the attractor. The goal is to learn the energy functional $V(w)$ such that the resulting ellipsoid is a tight outer estimate of the attractor. To this end, we include a regularization loss in the loss function that penalizes large ellipsoid volumes using Q (defined in Section 4.1) and a hyperparameter $\lambda > 0$.

$$\text{Loss} = \frac{1}{N} \sum_{i=1}^N \|w^*_{\text{next},i} - w_{\text{next},i}\|_2^2 + \lambda \frac{1}{\sqrt{\det(Q)}}, \tag{5}$$

## 5 Numerical Experiments

### 5.1 Lorenz 63

We first apply our methodology to the Lorenz 63 system (Lorenz, 1963), a classic low-dimensional model for visualizing chaotic dynamics and strange attractors. As shown in Figure 3a, our model generates a stable, 40,000-step trajectory from an unseen initial condition that accurately recovers the geometry of the true attractor. The learned ellipsoid provides a tight outer-estimate of this attractor, validating the effectiveness of volume regularization in learning a meaningful invariant set.

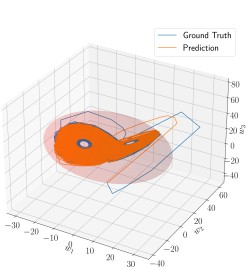 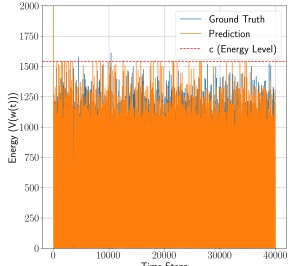 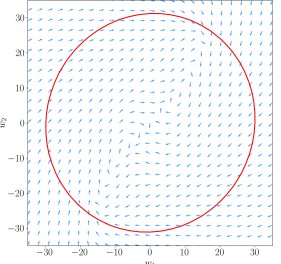

(a) Long-term trajectory rollout.     (b) Learned energy level $V(w_t)$.     (c) Learned flow field dynamics.

Figure 3: **Lorenz 63 prediction results.** (a) A 40,000-step trajectory generated by the model from an unseen initial condition (orange) compared to the ground-truth attractor (blue). The learned invariant ellipsoid (red) is a tight outer-estimate of the strange attractor. (b) The energy of the predicted trajectory quickly drops below the energy level $c$, and remains bounded by $c$. (c) The learned flow field on the $w_1$-$w_2$ plane, showing that vectors correctly point inwards across the ellipsoid boundary.

The learned energy function evaluations on both prediction and true trajectories are visualized in Figure 3b, which validates that the energy level of the true system is indeed bounded by the level set parameter $c$, and that the learned model is dissipative as it quickly loses energy and remains in the invariant set. Figure 3c further visualizes this dissipative behavior by showing the learned flow field on the $w_1 - w_2$ plane. The vector fields along the boundary of the learned ellipsoid all point inwards, ensuring that the set is positively invariant. Together, these results show that our framework successfully enforces long-term stability while learning an energy function that characterizes the system's attractor. The governing equations and implementation details are provided in Appendix C.1.

## 5.2 KURAMOTO–SIVASHINSKY

We next validate our framework on the chaotic one-dimensional Kuramoto–Sivashinsky (KS) equation (Kuramoto, 1978), a more challenging PDE benchmark. We compare our model (ECO), which adds the dissipative projection layer to a DeepONet backbone, against an unconstrained vanilla DeepONet. The results in Figure 4 highlight the critical role of the projection layer. When rolled out for 2000 time steps from an unseen initial condition, the unconstrained DeepONet quickly becomes unstable and its predictions blow up. In contrast, our projected model remains bounded and successfully recovers the complex spatio-temporal flow patterns of the true dynamics (Figure 4a).

Our method not only ensures boundedness but also successfully recovers the system's invariant statistics. By projecting the trajectories onto their first two principal components, Figure 4b shows that our model's predictions exhibit a similar distribution as the ground-truth data, demonstrating our model's capability to reconstruct invariant statistics on the strange attractor. The unconstrained model, prior to divergence, samples a sparse and unstructured distribution. Furthermore, the learned energy level (red ellipse) effectively bounds the attractor. We provide additional statistical property evaluations in Appendix C.2 to further validate these findings, along with implementation details.

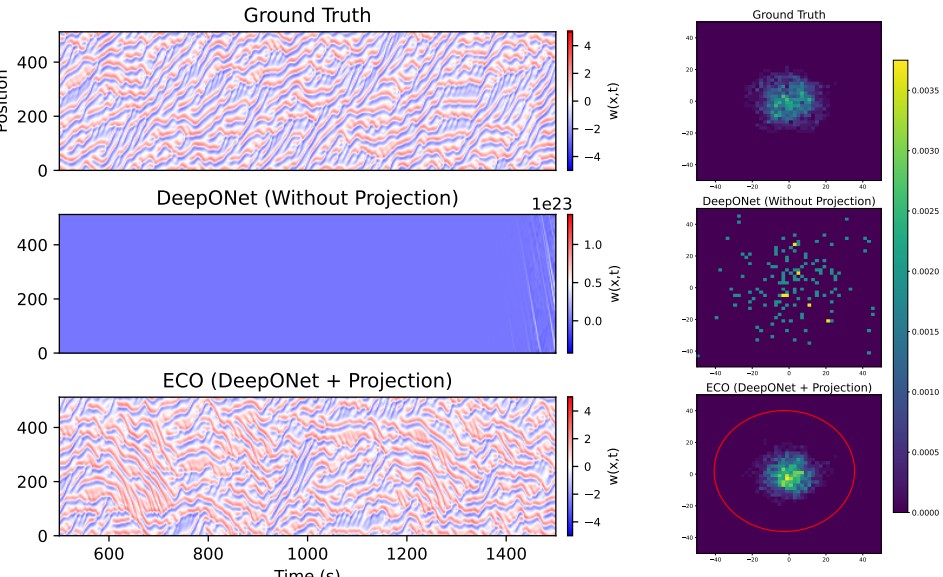

(a) Spatio-temporal plots of trajectory rollout.  (b) Projection of trajectories onto PCA modes.

Figure 4: **KS prediction results.** (a) Comparison of ground-truth trajectory with predictions from vanilla and projected model. Trajectories are visualized for 1000 seconds after a 500-second transition. The vanilla model blows up, while the projected model stays bounded. (b) Projection of trajectories onto the first two PCA modes. The projected model and ground-truth sample the strange attractor. The red line represents the learned invariant set projected onto the first two PCA modes.

## 5.3 NAVIER–STOKES

Finally, we test our framework on the two-dimensional Navier–Stokes equations (Temam, 2024) with Kolmogorov forcing, a challenging benchmark for chaotic, high-dimensional PDEs. We compare the long-term statistical properties of our model (ECO), against an unconstrained baseline DeepONet and the ground truth.

When rolled out autoregressively from an unseen initial condition for 10,000 time steps, the baseline DeepONet quickly grows to large values deviating from the attractor, as shown in Figure 5a. In contrast, our model (ECO) ensures trajectory boundedness near the attractor, hence generates a flow profile that closely resembles the patterns seen in the snapshots of the ground-truth trajectory. Furthermore, the stability allows our model to accurately capture the system's underlying statistical structure. As shown in Figure 5b, when trajectories are projected onto the first two principal components, our model's predictions correctly reproduce the distinct ring-shaped geometry of the true attractor, while the unconstrained model fails to do so. Implementation details and further statis-

tical analyses, including comparisons of the learned energy behavior, Fourier spectrum and spatial correlations, are provided in Appendix C.3.

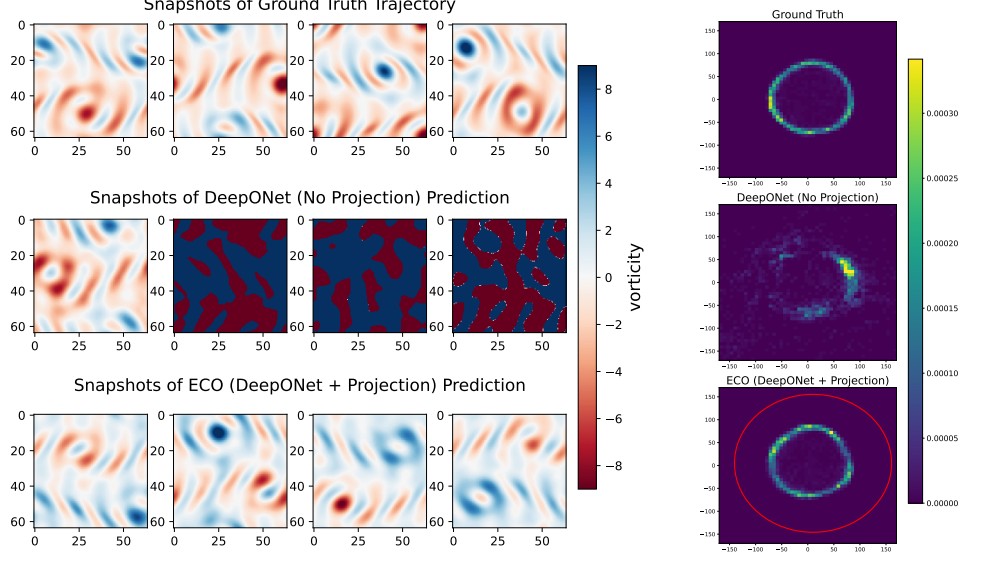

(a) Snapshots of NS flow behavior  (b) Projection of trajectories onto PCA modes.

Figure 5: **NS prediction results.** (a) Snapshots of the flow at various time steps for the model-predicted and ground-truth Navier–Stokes trajectories. The ground-truth and model-predicted dynamics exhibit similar patterns. (b) Projection of trajectories onto the first two PCA modes. The predicted dynamics capture the ring-shaped structure of the attractor.

Table 1: Quantitative comparison of long-term statistical accuracy. ECO with a dissipative projection layer consistently outperforms the unconstrained baseline across both dynamical systems.

| System | Approach | KL Divergence (physical) | KL Divergence (PCA) | Log-Spectral Distance |
|---|---|---|---|---|
| Kuramoto–Sivashinsky | DeepONet (Baseline) | 0.6268 | 11.88 | 33.14 |
| | **ECO (Projected)** | **0.0208** | **1.268** | **0.0186** |
| Navier–Stokes | DeepONet (Baseline) | 0.141 | 5.298 | 11.79 |
| | **ECO (Projected)** | **0.06221** | **0.9877** | **0.3689** |

To quantitatively validate our approach, we compare the statistical accuracy of ECO against the unconstrained baseline on the two PDE benchmarks. As shown in Table 1, our method significantly improves key statistical metrics like the KL divergence and log-spectral distance (detailed in Appendix B). It is noteworthy that these gains are achieved without providing any prior information about the system's dynamics or its attractor, as our model learns the stabilizing energy function and the dynamics purely from data.

## 6 CONCLUSION

To address the fundamental challenge of long-term stability in data-driven chaotic dynamics models, we introduced ECO, an energy-constrained operator design. Our framework jointly learns the system dynamics and a quadratic energy functional, and proposes a novel convex quadratic projection layer to enforce a computationally efficient dissipativity condition. This approach, to our knowledge, is the first to provide explicit theoretical guarantees on the long-term boundedness of a learned chaotic emulator. We demonstrated ECO's effectiveness on benchmarks including Lorenz 63, Kuramoto–Sivashinsky, and Navier–Stokes, where it produced stable, long-horizon forecasts that accurately reproduced the systems' invariant statistics while unconstrained baselines failed. This work demonstrates the value of building physically constrained models, marking a step toward more reliable and robust scientific machine learning.

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

## A    PROOF FOR THEORETICAL RESULTS

*Proof for Proposition 1.* By definition, condition (ii) implies that $M(c)$ is indeed a positively invariant set. Since $V$ is radially unbounded, for any $\alpha > 0$, we can find $r_\alpha$ such that $V(w) > \alpha$ for all $\|w\| > r_\alpha$. Therefore, any level set of $V$ is bounded as $\{x : V(w) \le \alpha\} \subset B(r_\alpha)$. Thus, $M(c)$ is both positively invariant and bounded.

Based on the positive invariance property, any trajectory starting with $w_0 \in M(c)$ will always stay within $M(c)$, meaning that $\lim_{t \to \infty} \text{dist}(w_t, M(c)) = 0$ is satisfied.

Consider a trajectory $\{w'_t\}_{t \in \mathbb{N}}$ that starts outside of the level set, i.e. $w'_0 \notin M(c)$ and $V(w'_0) > c > 0$. Suppose the trajectory never enters $M(c)$, i.e., $\forall t \in \mathbb{N}, V(w'_t) > c$. Using condition (i), we have $V(w'_{t+1}) \le \alpha V(w'_t)$, which implies $V(w'_t) \le \alpha^t V(w'_0)$. For any $t \ge \log_\alpha \left( \frac{c}{V(w'_0)} \right)$, $V(w'_t) \le c$ which contradicts the prior assumption. In fact, the trajectory $\{w'_t\}_{t \in \mathbb{N}}$ will enter $M(c)$ in finite time at most $\log_\alpha \left( \frac{c}{V(w'_0)} \right)$ steps.   $\square$

*Proof for Lemma 1.* Let $\langle u, v \rangle_Q := u^\top Q v$ and $\|u\|_Q := \sqrt{u^\top Q u}$ so that $V(w) = \|w - w_c\|_Q^2$. By construction,

$$\bar{w} - w_c = \sqrt{b}\,(L^\top)^{-1} \frac{\hat{w} - w_c}{\|\hat{w} - w_c\|_2} \implies V(\bar{w}) = \|\bar{w} - w_c\|_Q^2 = b$$

Define $\hat{v} := \hat{w} - w_c$, $\bar{v} := \bar{w} - w_c$, and $\gamma := \sigma(k(b - V(\hat{w}))) \in (0, 1)$. Then

$$w^* - w_c = \gamma \hat{v} + (1 - \gamma)\bar{v}.$$

By Minkowski's inequality for the $Q$-norm,

$$\|w^* - w_c\|_Q \le \gamma \|\hat{v}\|_Q + (1 - \gamma)\|\bar{v}\|_Q = \gamma\sqrt{V(\hat{w})} + (1 - \gamma)\sqrt{b}.$$

Hence

$$V(w^*) \le \left( \sqrt{b} + \gamma(\sqrt{V(\hat{w})} - \sqrt{b}) \right)^2. \tag{6}$$

If $V(\hat{w}) \le b$, then the right-hand side of (6) is at most $b$. So it suffices to consider $V(\hat{w}) \ge b$.

Let $s = \sqrt{V(\hat{w})} \ge \sqrt{b}$ and $t = s - \sqrt{b} \ge 0$. Then

$$\gamma = \frac{1}{1 + e^{k(s^2 - b)}} \le \frac{1}{2 + k(s^2 - b)}$$

since $e^x \ge 1 + x$ for all $x \ge 0$. Note that $s^2 - b = t(t + 2\sqrt{b})$, so

$$\gamma t \le h(t) := \frac{t}{2 + k(t^2 + 2\sqrt{b}\,t)}.$$

Since the derivative vanishes at $t^\star = \sqrt{2/k}$, let $\delta = (2kb + 2\sqrt{2kb})^{-1}$.

$$h'(t) = \frac{2 - kt^2}{(2 + k(t^2 + 2\sqrt{b}\,t))^2}, \implies \max_{t \ge 0} h(t) = \frac{\sqrt{2/k}}{4 + 2\sqrt{2kb}} = \frac{\sqrt{b}}{2kb + 2\sqrt{2kb}} = \delta\sqrt{b}.$$

Thus, for all $s \ge \sqrt{b}$,

$$\gamma(\sqrt{V(\hat{w})} - \sqrt{b}) \le \delta\sqrt{b}. \tag{7}$$

Combining (6) and (7),

$$V(w^*) \le \left( \sqrt{b} + \delta\sqrt{b} \right)^2 = (1 + \delta)^2 b.$$

$\square$

*Proof for Theorem 1.* Let $b = \alpha[V(w_t) + \text{ReLU}(c - V(w_t))]$, it directly follows from $c > 1/\alpha$ that $b \geq \alpha c > 1$. The inequality constraint in (3) states that $V(w_{t+1}) \leq b$. Since $b > 1$, we have $(2kb + 2\sqrt{2kb})^{-1} < (2k + 2\sqrt{2k})^{-1}$. Using Lemma 1, the final output of our model $w_{t+1}^*$ satisfies that $V(w_{t+1}^*) \leq (1 + (2k + 2\sqrt{2k})^{-1})^2 b$. Since $\alpha < [1 + (2k + 2\sqrt{2k})^{-1}]^{-2}$, we can always find some $0 < \alpha_0 < 1$ such that $[1 + (2k + 2\sqrt{2k})^{-1}]^2 \alpha \leq \alpha_0 < 1$. Consequently, the inequality constraint in (3) is satisfied:

$$V(w_{t+1}^*) \leq (1 + (2k + 2\sqrt{2k})^{-1})^2 \alpha[V(w_t) + \text{ReLU}(c - V(w_t))] \leq \alpha_0[V(w_t) + \text{ReLU}(c - V(w_t))]$$

Given the equivalency between the constraint (3) and the conditions in Proposition 1, we have now shown the learned dynamics $w_{t+1}^* = G^*(w_t^*)$ is indeed dissipative.

Note that for the predicted trajectory under the learned dynamics, at any time $t > 0$, $V(w_t^*)$ is strictly bounded by $\max\{V(w_0), c\}$, which implies $V(w_t^*) \leq V(w_0)$. Let $z = w_t^* - w_c$, the constraint can be rewritten as $z^T Q z \leq V(w_0)$. Since $Q$ is positive definite, we have $\lambda_{\min}(Q)\|z\|^2 \leq z^T Q z \leq V(w_0)$. Thus $w_t^*$ will always be bounded with respect to its $L^2$ norm. $\square$

# B    METRICS FOR STATISTICAL PROPERTY EVALUATION

We use three statistical metrics to evaluate the effectiveness of our model in capturing statistical properties.

**KL Divergence (physical):** We compare the pixel-wise distribution of velocity (Kuramoto–Sivashinsky) or vorticity (Navier–Stokes) between the ground-truth data and the model-predicted trajectories. We evaluate the distance between distributions with the Kullback–Leibler divergence:

$$D_{KL}(P||Q) = \sum_{x \in \mathcal{X}} P(x)\log\frac{P(x)}{Q(x)} \tag{8}$$

In this case, $x \in \mathcal{X} \subset \mathbb{R}$ is the support of the physical quantity (velocity or vorticity) across all pixels in space and time. That is, a single grid point on the spatial domain at a single snapshot in time represents one sample from the distribution. We use $P(x)$ as the ground-truth distribution and $Q(x)$ as the predicted distribution.

**KL Divergence (PCA):** We also compare the Kullback–Leibler divergence between the distributions of the physical quantities onto their two PCA modes. Projecting every snapshot in time onto the first two PCA modes gives a 2D distribution of the trajectories in PCA space, as seen in Figures 4 and 5b. We then compute the KL divergence from Equation 8, where $x \in \mathcal{X} \subset \mathbb{R}^2$ is the support of the projected snapshot onto a 2D point across all snapshots in time. We use $P(x)$ as the ground-truth distribution and $Q(x)$ as the predicted distribution.

**Log-Spectral Distance:** We create energy spectra for predicted trajectories by performing a discrete Fourier transform in the spatial domain of each snapshot in time. The energy of a given Fourier mode is computed by taking the average in time of the square signal. The Log-Spectral Distance between the ground-truth spectrum and the predicted spectrum is defined as

$$D_{LS} = \left\{\frac{1}{N}\sum_{n=1}^{N}\left[\log P(n) - \log\hat{P}(n)\right]^p\right\}^{1/p} \tag{9}$$

Here, $P(n)$ is the ground-truth energy of the $n$th Fourier mode and $\hat{P}(n)$ is the predicted energy of the $n$th Fourier mode. We use $p = 2$ for the results reported in Table 1.

# C  IMPLEMENTATION DETAILS AND ADDITIONAL NUMERICAL RESULTS

## C.1  LORENZ 63

**Governing Equations.** The Lorenz 63 system is described by the following ordinary differential equations (ODEs):

$$\dot{w}_1 = \sigma(w_2 - w_1), \ \dot{w}_2 = w_1(\rho - w_3) - w_2, \ \dot{w}_3 = w_1 w_2 - \beta w_3.$$

where $w \in \mathbb{R}^3$ is the system state. We use parameters $\sigma = 10.0, \rho = 28.0, \beta = 8/3$ which generate chaotic behaviors.

**Dataset Generation.** The training data was generated from a single long trajectory integrated for 10,000 seconds with a sampling interval of 0.05 seconds. The trajectory was initialized at a random state outside the strange attractor to ensure the model learns the dissipative flow. For evaluation, we generated a test trajectory by starting from a new, unseen initial condition for 2,000 seconds.

**Model Architecture.** This system can be viewed as a special case of an infinite-dimensional dynamical system where the spatial domain consists of just three discrete points. In this simplified context, backbone neural operators like the Fourier Neural Operator (Kovachki et al., 2023) and DeepONet (Lu et al., 2021) reduce to a simple multilayer perceptron (MLP). Accordingly, we constructed our unconstrained neural operator emulator, $\hat{G}$, as a feedforward neural network with 6 hidden layers, each containing 150 neurons.

## C.2  KURAMOTO–SIVASHINSKY

**Governing Equations.**

$$
\begin{aligned}
w_t + w_{xx} + w_{xxxx} + \frac{1}{2}(w^2)_x &= 0, \quad (t,x) \in [0,\infty) \times [0,L] \\
w(0,x) &= w^0(x), \quad x \in [0,L]
\end{aligned}
\tag{10}
$$

We use a domain length $L = 32\pi$ to generate chaotic behavior.

**Dataset Generation.** The training dataset consists of six trajectories simulated for 500 seconds, with snapshots saved every 1 second at a spatial resolution of 512 points. The validation set contains two trajectories with the same discretization. All trajectories were initialized with random conditions.

**Model Architecture.** We construct two models based on DeepONet backbones, one with the added projection layer discussed in Section 4.2 and one vanilla model without. For both models, the DeepONet branch network consists of three convolutional layers of output dimension 32, 64, 128, and two fully connected layers, each with 256 neurons. The trunk network consists of four fully connected layers, each with 256 neurons.

**Additional Results.** To further validate our approach, we provide a detailed comparison of the learned dynamics. Figure 7 visualizes the learned energy function behavior and Fourier spectrum of the predicted trajectories, while Figure 6 examines the statistical distribution constructed by the predicted trajectories. Both figures confirm that our projected model (ECO) remains bounded and accurately captures the invariant statistics of the true system.

## C.3  NAVIER–STOKES

**Governing Equations.** The 2D Navier–Stokes equations in vorticity form with Kolmogorov forcing are given by:

$$
\begin{aligned}
w_t &= -u \cdot \nabla w + \frac{1}{Re}\nabla^2 w - n\cos(ny), \quad (t,x,y) \in [0,\infty) \times [0,2\pi] \times [0,2\pi] \\
w(0,x,y) &= w^0(x,y), \quad (x,y) \in [0,2\pi] \times [0,2\pi]
\end{aligned}
\tag{11}
$$

We use a wave number of the forcing function $n = 4$ and a Reynolds number of $Re = 40$.

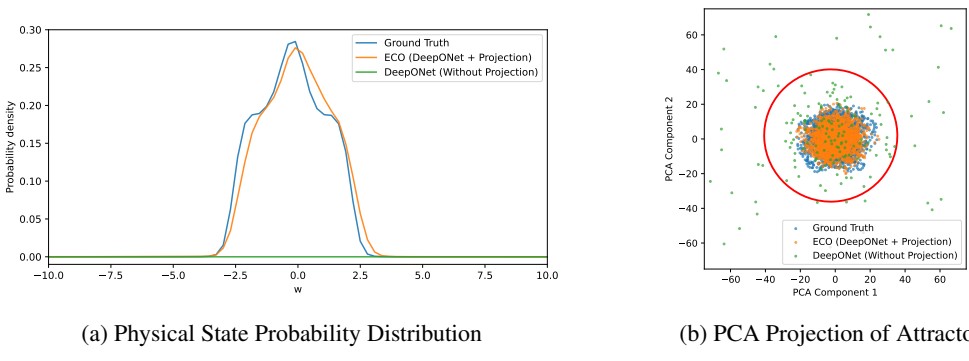

(a) Physical State Probability Distribution

(b) PCA Projection of Attractor

Figure 6: **KS: State-space and statistical distribution analysis.** (a) The probability distribution of the physical state variable for our projected model closely matches the ground truth, while the unconstrained model's distribution is nearly flat due to its divergence. (b) Trajectories projected onto the first two PCA modes confirm that ECO's predictions correctly sample the attractor's geometry and the learned invariant ellipsoid (red) provides a tight outer-estimate of the true attractor.

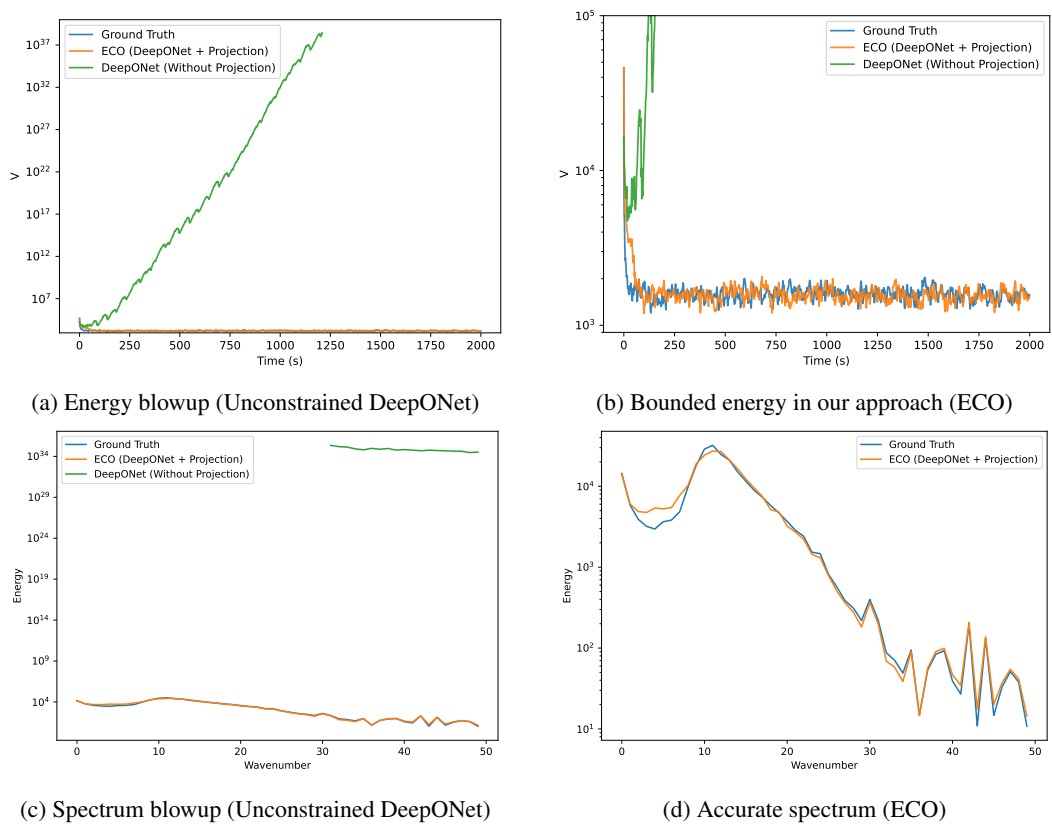

(a) Energy blowup (Unconstrained DeepONet)

(b) Bounded energy in our approach (ECO)

(c) Spectrum blowup (Unconstrained DeepONet)

(d) Accurate spectrum (ECO)

Figure 7: **KS: Energy and Fourier spectrum comparison.** The top row shows the learned energy $V(w)$ over time, with the unconstrained model energy growing unbounded in a zoomed-out plot (a), while our model remains bounded and produces the same energy level as the ground truth (b). The bottom row shows the Fourier power spectrum, the instability of the baseline (c) versus the accurate spectrum reconstruction of our model with projection (d).

**Dataset Generation.**     The training dataset consists of 160 trajectories, each simulated for 500 seconds with snapshots saved every 1 second. Each snapshot has a spatial resolution of $64 \times 64$. The validation set contains 40 trajectories with the same discretization.

**Model Architecture.**  The DeepONet backbone consists of a branch network with four convolutional layers (output dimensions 64, 128, 256, 512) and two fully connected layers (1024 neurons each). The trunk network consists of five fully connected layers (1024 neurons each).

**Additional Results.**  To further validate our approach, we provide a detailed comparison of the learned dynamics. The results confirm that our projected model (ECO) remains bounded and accurately captures the invariant statistics of the true system, in contrast to the unconstrained baseline.

Figure 8 shows that while the unconstrained model's energy and Fourier spectrum blow up, ECO's predictions remain bounded and track the ground truth. This stability allows our model to learn the correct underlying statistics while the unconstrained model produces a sparse distribution, as shown in Figure 9. The probability distribution of the predicted state matches the true dynamics, and the trajectories correctly sample the attractor's geometry within the learned invariant ellipsoid. Finally, qualitative comparisons in Figure 10 show that ECOreproduces the correct large-scale spatial structures induced by the system's forcing.

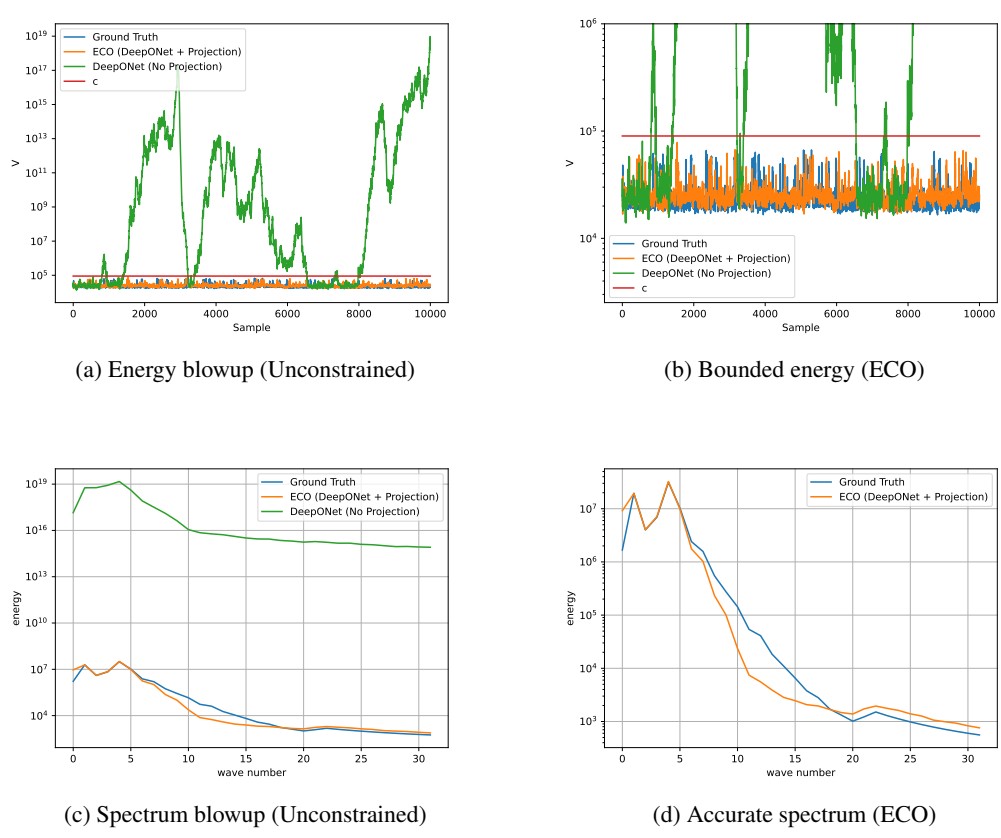

(a) Energy blowup (Unconstrained)  (b) Bounded energy (ECO)

(c) Spectrum blowup (Unconstrained)  (d) Accurate spectrum (ECO)

Figure 8: **NS: Energy and Fourier spectrum comparison.** The top row shows the learned energy over time, with the unconstrained model blowing up (a) while our model remains bounded (b). The bottom row shows the energy spectrum, highlighting the instability of the baseline (c) versus the accuracy of our model (d).

## D  FUTURE WORK

Future work could explore more expressive non-quadratic energy functionals, potentially even parameterized by another neural network, and extend the projection framework to enforce other nonlinear physical constraints. Investigating the computational scaling of this approach to even larger-scale and realistic systems also remains a promising direction for building reliable and physically consistent dynamics emulators.

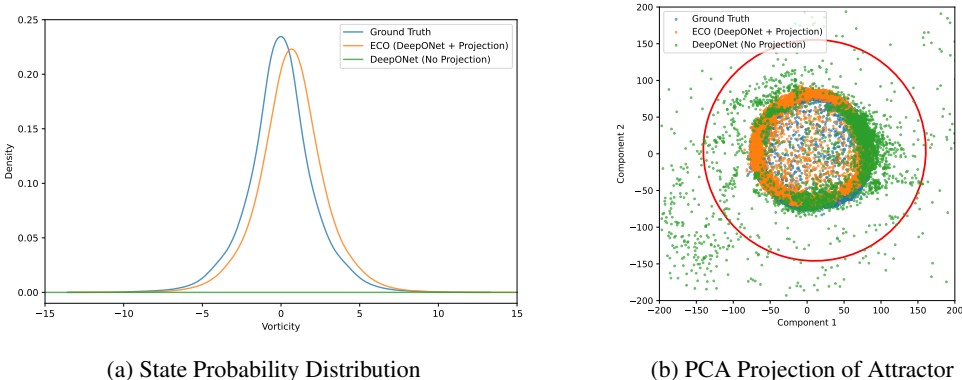

(a) State Probability Distribution      (b) PCA Projection of Attractor

Figure 9: **NS: State-space and statistical distribution analysis.** (a) The probability distribution of the state variable for our projected model matches the ground truth. (b) Trajectories projected onto the first two PCA modes confirm that ECO's predictions correctly sample the attractor's geometry within the learned invariant ellipsoid (red).

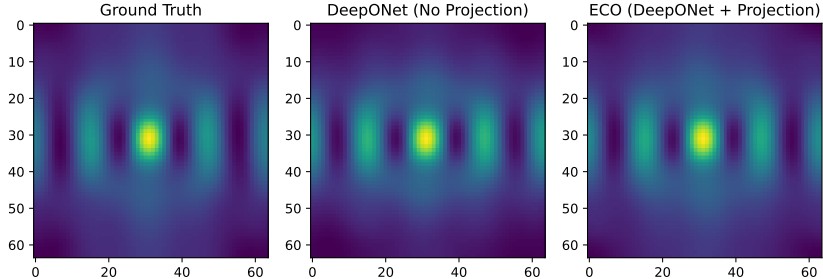

Figure 10: **NS: Spatial correlation of the ground-truth and predicted trajectories averaged in time.** The ground-truth and predicted dynamics show patterns consistent with the sinusoidal forcing with frequency 4.

## E LLM USAGE

We acknowledge the use of LLMs for assistance with polishing and language refinement. All content and ideas presented are the original work of the authors.

