# OpenReview forum: "ECO: Energy-Constrained Operator Learning for Chaotic Dynamics with Boundedness Guarantees"
_ICLR.cc/2026/Conference — ICLR 2026 Conference Withdrawn Submission_

### Official Review · Reviewer_Sx1L · 2025-10-26

**Soundness:** 2
**Presentation:** 2
**Contribution:** 2
**Rating:** 2
**Confidence:** 5

**Summary:**

## Summary
This paper introduces the Energy-Constrained Operator (ECO), a framework designed to learn the dynamics of dissipative chaotic systems while providing theoretical guarantees on the boundedness of long-term predictions. The core idea is to jointly learn the system's operator and a Lyapunov function with quadratic form. A novel projection layer is introduced to enforce a dissipativity condition derived from control theory, ensuring that all predicted trajectories converge to a learned, bounded invariant set guided by the learned Lyapunov function. The authors validate their method on the Lorenz 63, Kuramoto-Sivashinsky, and 2D Navier-Stokes equations, showing that ECO can produce stable long-horizon forecasts, unlike an unconstrained baseline model which suffers from trajectory blow-up.

**Strengths:**

## Strengths
- The paper addresses a critical and well-known challenge in data-driven modeling of chaotic systems: the instability and divergence of long-term autoregressive predictions. The goal of enforcing stability by construction is highly relevant and important for the field.
- The primary contribution is the theoretical, formal guarantees for the boundedness of the learned dynamics. Integrating Lyapunov stability theory into the neural operator architecture through a differentiable quadratic projection layer is elegant.
- The experiments clearly illustrate the failure of DeepONet and the success of the proposed ECO framework in maintaining stability and capturing the the attractor on challenging PDE benchmarks.

**Weaknesses:**

## Weaknesses
While the core idea is promising, the paper has several notable weaknesses.
- One weakness is the mismatch between the introduced Lyapunov energy-functional layer and the paper’s *operator-learning* framework. As stated in Theorem 1 (and its proof via bounds on the quadratic form), the energy layer and its guarantees are derived for a finite-dimensional system by spatial discretization. Specifically, the learned Lyapunov functional $V(w)=(w-w_c)^\top Q(w-w_c)$ acts on $w\in\mathbb{R}^n$ with the SPD matrix $Q\in\mathbb{R}^{n\times n}$, so both $V$ and its stability certificate depend on the chosen grid resolution. This conflicts with the discretization-independent goal of operator learning [Kovachki et al., 2023]: when the grid size $n$ changes, $Q$ and $w_c$ must be re-trained. The authors should state this limitation clearly. The paper would be much stronger if it discussed the challenge and potential solution of extending these guarantees to a true, discretization-independent function space setting.

- The experimental validation, while demonstrating the value of the projection layer, is limited to an ablation study comparing ECO (DeepONet + Projection) against the standard DeepONet baseline. DeepONet is a general-purpose model, not one specifically designed for the stable long-term prediction of chaotic systems. This comparison fails to position ECO within the broader context of surrogate models tailored for *chaotic dynamics*. At minimum, the authors should include one baseline explicitly designed for long-term stable rollout—e.g., the Markov Neural Operator (MNO; Li et al., 2022), which the authors already cite in the manuscript and for which default hyperparameters and a public codebase enable reproducible comparisons. The authors do not provide reasons for omitting such baselines. Without the comparison, it is unclear whether ECO offers advantages over other surrogates designed for chaotic systems.

- The literature review overlooks recent advances in two areas: (1) surrogate modeling for chaotic systems (e.g., Schiff et al.; 2024, Cheng et al., 2025; He et al., 2025; Brenner, et al., 2025) and (2) Lyapunov/energy-based stabilization of forecasts (e.g., Rodriguez et al., 2022). A more thorough review is needed to contextualize and narrow the novelty and advantages of this work.

- The provided repository is appreciated, but there is no instructions. Please add a reproducibility guide to reproduce the reported results.

**Questions:**

## Questions
- Why does the DeepONet baseline in Fig. 4 fail immediately ($\approx t=0$)? Under standard settings, one would expect at least a few reasonable rollout steps. Could the author include a zoomed view of the first few steps or justify this unexpected result.

- Following Weakness #2, could the authors justify the decision to compare only against a standard DeepONet baseline? Why were other stability-focused methods (e.g., MNO) not included in the comparison?

- The choice of a quadratic Lyapunov function is key to deriving the energy layer and its theoretical guarateen. What are the limitations of this choice? How would the framework perform on systems where the strange attractor's basin is not well-approximated by an ellipsoid? Does this choice restrict the types of dynamics ECO can stably learn?

- The paper's problem formulation, theoretical guarantees, and experiments all focus on autonomous (time-indepdent) dynamical systems. Could the author comment on how would the proposed ECO framework and its theoretical guarantees be extended to handle non-autonomous systems, where the governing PDE is time-depdent?

- The paper notes that for high-dimensional outputs, a diagonal $Q$ matrix is used for efficient computation. Could the author comment on (1) The diagonal simplification means the learned $V(x)$ must be axis-aligned. How does this diagonal simplification limit the model's ability to learn a complex attractor, which is unlikely to be axis-aligned in the chosen coordinates? (2) How the grid resoultion affects the training and performance of the learnable matrix $Q$? For example, does training on a finer grid generally lead to better forecasting accuracy?

## References
- Rodriguez, Ivan Dario Jimenez, Aaron Ames, and Yisong Yue. "Lyanet: A lyapunov framework for training neural odes." International conference on machine learning. PMLR, (2022).
- Kovachki, Nikola, et al. "Neural operator: Learning maps between function spaces with applications to pdes." Journal of Machine Learning Research 24.89 (2023): 1-97.
- Li, Zongyi, et al. "Learning chaotic dynamics in dissipative systems." Advances in Neural Information Processing Systems 35 (2022): 16768-16781.
- Schiff, Yair, et al. "Dyslim: Dynamics stable learning by invariant measure for chaotic systems." arXiv preprint arXiv:2402.04467 (2024).
- Cheng, Xiaoyuan, et al. "Learning chaos in a linear way." arXiv preprint arXiv:2503.14702 (2025).
- He, Yi, et al., Chaos Meets Attention: Transformers for Large-Scale Dynamical Prediction. Proceedings of the 42nd International Conference on Machine Learning (2025).
- Brenner, Manuel, et al. "Learning Interpretable Hierarchical Dynamical Systems Models from Time Series Data." The Thirteenth International Conference on Learning Representations (2025).

---

> ### Author Response · Authors · 2025-11-13
>
> > Strengths: The paper addresses a critical and well-known challenge …
>
> > The primary contribution is the theoretical, formal guarantees …
>
> > The experiments clearly illustrate the failure of DeepONet and the success of the proposed ECO framework…
>
> Thank you for acknowledging our key contributions in many aspects. We are not sure why this is not reflected in the reviewer’s rating.
>
>
> > One weakness is the mismatch …  extending these guarantees to a true, discretization-independent function space setting
>
> Parameterizing a network for a positive definite $q(x)$ can be directly incorporated in our framework to approximate an integral energy function $V(w) = \int q(x)w(x)^2dx$, which is an interesting and immediate extension of this work that we have been considering. The quadratic form of this integral approximation would clearly be amenable to our proposed quadratic projection.
> > The experimental validation …whether ECO offers advantages over other surrogates designed for chaotic systems
>
> We emphasize that the advantage ECO offers over other surrogates designed for chaotic systems lies in its boundedness guarantees. Other works lack theoretical results demonstrating guarantees or rely on specific network parameterizations to achieve stability. The experimental validation serves to demonstrate that we can achieve boundedness guarantees without sacrificing model expressivity. Our projection layer is model-agnostic and could therefore be incorporated into the models tailored to chaotic dynamics. On the other hand, MNO uses system-specific prior information to hard-code their projection. Our method removes the requirement for such critical information and discovers energy dissipation constraints from data. Therefore, it would not be a fair or meaningful comparison.
>
> > Question 1
>
> The DeepONet baseline does not fail immediately, and we agree that would be an unexpected result. Figure 4 shows $T = 500$s to $T = 1500$s. We refer the interested reader to Figure 7b in Appendix C, which shows that the unconstrained model produces trajectories with energy levels consistent with the projected model for the initial time steps, before becoming unbounded.
>
> > Question 2
>
> Please see response to Weakness #2.
>
> > Question 3
>
> As addressed in sections 2 and 3, the shape of the strange attractor is system-dependent and can be intractable to characterize, and therefore is likely not well-approximated by an ellipsoid. Our approach does not aim to approximate the attractor by an ellipsoid, but rather to learn an outer bound surrounding the attractor that enforces boundedness, relying on the expressive power of the network to capture the true shape of the attractor. The experimental section, specifically Figure 3a, gives an example of this, showing that the ellipsoid itself does not serve as the attractor shape, but an outer bound within which the model can traverse the true attractor. For stability guarantees, the Lyapunov function must be positive definite and radially unbounded (see Proposition 1), and a quadratic function is clearly suitable for these requirements, regardless of the underlying dynamics.
>
> > Question 4
>
> ECO’s boundedness guarantees hold regardless of the model backbone, so a model backbone that takes time as an input would not change the theoretical guarantees. For time-dependent dynamics, the statistical objective we (and others) target in this work presupposes time-invariant dynamics. In time-dependent systems where the attractor evolves in time, the notion of ergodic invariant statistics may not apply in the same form. Characterizing time-dependent attractors would therefore require a reformulation of the learning objective, which could be an interesting future direction but is outside the scope of this work.
>
> > Question 5
>
> The shape of the learned invariant set, which in this case is an axis-aligned ellipsoid, is not intended to represent the true shape of the attractor, but rather bound predictions such that the model can learn the true attractor.

---

### Official Review · Reviewer_Pn8K · 2025-10-27

**Soundness:** 3
**Presentation:** 3
**Contribution:** 2
**Rating:** 4
**Confidence:** 3

**Summary:**

The paper introduces ECO (Energy-Constrained Operator) learning, a framework designed to ensure long-term stability in the learning of chaotic dynamical systems. The method augments a neural operator (implemented via DeepONet) with a learnable quadratic Lyapunov energy function and a convex quadratic projection layer that enforces a discrete-time dissipativity condition. This combination yields a theoretical boundedness guarantee for the learned dynamics. Experiments on Lorenz-63, Kuramoto–Sivashinsky, and 2D Navier–Stokes systems demonstrate empirically stable rollouts and reduced divergence compared to an unconstrained baseline.
While the framework is clearly presented and mathematically coherent, most of its core ideas have strong precedents in existing literature on dissipative-Hamiltonian and Lyapunov-stable neural networks, as well as differentiable optimization layers. ECO’s novelty lies mainly in its discrete-time treatment and the simplicity of its closed-form constraint enforcement.

**Strengths:**

Provides a clear and formal discrete-time boundedness guarantee.

Integrates the constraint enforcement in a simple, computationally efficient closed-form layer.

Demonstrates convincing empirical prevention of trajectory blow-up on canonical chaotic systems.

Theoretical results are stated cleanly and tied directly to the implementation.

**Weaknesses:**

The novelty is limited—energy-based constraints, Lyapunov guarantees, and differentiable convex projections have been previously  studied.

Empirical baselines are thin (only DeepONet). Comparisons to FNO, reservoir computing, or dissipativity-aware models would greatly strengthen the manuscript.

Overclaims originality (“first with guarantees for chaotic dynamics”)—this should be softened and scoped precisely.

No ablations on key hyperparameters (k, alpha,Q) or quantification of how tightly the learned ellipsoid captures the attractor.

**Questions:**

Does the boundedness guarantee still hold if Q is learned as non-diagonal?

How sensitive is the approach to the choice of energy level c?

---

> ### Author Response · Authors · 2025-11-13
>
> > Provides a clear and formal discrete-time boundedness guarantee.
>
> > Integrates the constraint enforcement in a simple, computationally efficient closed-form layer.
>
> > Demonstrates convincing empirical prevention of trajectory blow-up on canonical chaotic systems.
>
> > Theoretical results are stated cleanly and tied directly to the implementation.
>
> Thank you for acknowledging our key contributions in many aspects. We are not sure why this is not reflected in the reviewer’s rating.
>
> > The novelty is limited – energy-based constraints, Lyapunov guarantees, and differentiable convex projections have been previously studied.
>
> While our approach shares a conceptual connection with works regarding energy-based constraints, Lyapunov stability analysis, and differentiable convex projections, we emphasize that ECO is fundamentally different both in formulation and purpose. ECO introduces a closed-form, differentiable projection layer for quadratic constraints, derived from Lyapunov-inspired dissipative conditions, enabling exact and efficient enforcement of boundedness with any neural operator. Our approach does not rely on system-specific parameterization, iterative approaches, or regularization-based soft constraint methods for enforcing dissipativity, yielding a novel framework that provides formal, explicit boundedness guarantees for data-driven chaotic PDE learning.
>
>
> > Empirical baselines are thin (only DeepONet). Comparisons to FNO, reservoir computing, or dissipativity-aware models would greatly strengthen the manuscript.
>
> This work addresses a different axis of comparison with prior works: the lack of theoretical guarantees in existing data-driven models for chaotic dynamics. While other methods show strong empirical success in predicting the trajectories of chaotic systems, many require specific parameterizations and/or lack stability assurances. Our experimental setup was to demonstrate the ability of our framework to provide boundedness guarantees that allow the expressive power of the backbone operator (in this case, the DeepONet) to capture the dynamics. The proposed layer is architecture-agnostic and can, in principle, be applied to any operator-learning model, including those mentioned by the reviewer.
>
> > No ablations on key hyperparameters (k, alpha,Q)
>
> We discuss k and alpha in section 4.3 (see “Practical Implications”). The values of k and alpha are pertinent to the theoretical stability guarantees. Once they are selected such that they meet the requirements detailed in this section, they do not require additional tuning.
> We provide results showing the ellipsoid defined by learnable Q (or a lower-dimensional projection of it) in the results section (see Figures 3a, 4b, and 5b). Here, we empirically show that the ellipsoid is a sufficient outer bound on the attractor, providing boundedness guarantees, while simultaneously allowing the expressive power of the DeepONet backbone to capture the dynamics and therefore the complex shape of the attractor.
>
>
> > Does the boundedness guarantee still hold if Q is learned as non-diagonal?
>
> Yes. The proof in Appendix A does not rely on diagonal Q. The lower-dimensional illustrative Lorenz 63 experiments use a non-diagonal Q.

---

### Official Review · Reviewer_3zfM · 2025-10-30

**Soundness:** 2
**Presentation:** 2
**Contribution:** 2
**Rating:** 2
**Confidence:** 3

**Summary:**

This paper introduced "Energy-Constrained Operator" leverage concepts from control theory to develop algebraic conditions
based on a learnable energy function. ECO integrates control theory principles into machine learning, using a learnable Lyapunov-based energy function and a convex quadratic projection layer to guarantee that model predictions remain bounded.  The paper performed on some chaotic equations like Lorenz63/KS/kolmogrov flows.

**Strengths:**

Good story on involving the control theory to help ML model

**Weaknesses:**

Despite good theoretic insights, the results doesn't look promising. Especially, I am not sure if the DeepOnet is tuned properly. No ML model parameters is provided. Also, wether the authors have checked MNO results. The PDF of the data looks shifted from the ground truth making the results less convincing.

**Questions:**

1. What's the total parameter sets?
2. Have you compared MNO results from Li et. al?

---

> ### Author Response · Authors · 2025-11-13
>
> > Good story on involving the control theory to help ML model
>
> This is an inaccurate summary of our contributions; see the end of the introduction.
>
> > Despite good theoretic insights, the results doesn't look promising … less convincing.
>
> The goal of our paper is to provide a framework for neural operators that establishes formal trajectory boundedness guarantees. Importantly, we develop a provably bounded neural operator, which the reviewer appears to overlook.
>
> The review comments are factually incorrect. The DeepONet baseline has been tuned carefully. The training dataset size and the architecture used for every numerical experiment have been reported in the paper: see Appendix C.
>
> > Question 1
>
> The architecture details are reported, including specific configurations of layers.
>
> > Question 2
>
> No. MNO uses system-specific prior information to hard-code their projection. Our method removes the requirement for such critical information and discovers energy dissipation constraints from data. Therefore, it would not be a fair or meaningful comparison.

---

### Official Review · Reviewer_vKse · 2025-10-31

**Soundness:** 3
**Presentation:** 1
**Contribution:** 2
**Rating:** 4
**Confidence:** 4

**Summary:**

The paper introduces the  Energy-Constrained Operator (ECO) , a neural operator for learning chaotic dynamics with formal boundedness guarantees. By integrating control-theoretic Lyapunov energy functions and a differentiable convex quadratic projection layer, ECO enforces dissipativity to ensure trajectory stability. Empirical results on Lorenz-63, Kuramoto–Sivashinsky, and Navier–Stokes (NS) systems demonstrate stable long-horizon forecasts and accurate recovery of invariant statistics, outperforming unconstrained DeepONet in chaotic PDE modeling.

**Strengths:**

The paper has the following strengths worth credits:

1. Integrating control-theoretic Lyapunov energy functions to enforce dissipativity in learning chaotic systems.
2. The development of a convex quadratic projection layer to generate bounded predictions.
3. Appendix B provides meaningful metrics for evaluating long-term forecasting performance in chaotic systems.

**Weaknesses:**

The paper should be improved considering the following facts:

1. The novelty of ellipsoidal constraints is **incremental** compared with Markov neural operator (MNO). ECO replaces the spherical hard constraint with ellipsoidal constraints and introduces a differentiable projection layer.
2. Experiments and evaluations are limited to compare ECO with DeepONet. In Figure 4-5, DeepONet is not a strong baseline.
Leading methods such as MNO (a constrained neural operator method), Poincaré Flow Neural Networks (PFNN) (a constrained koopman operator method), Mamba (a state space model) are not addressed. This makes it difficult to assess the effectiveness of ECO compared with existing constrained methods for chaotic systems.
3. The Ablation study is insufficient. The paper only studied on with/without projection in table 1. It would be more convincing to include ablation studies on the effect of different components in ECO, such as the choice of hyperparameters, weighing regularizers and model sensitivity to them, with/without energy function, etc.
4. Though the paper provides several metrics for evaluating long-term forecasting performance in chaotic systems in Appendix B, many established metrics in the literature are not discussed, such as Wasserstein Distance, Maximum Mean Discrepancy (MMD), and Lyapunov Exponent, Lyapunov time. Including these metrics would enhance the comprehensiveness of the evaluation.

**Questions:**

1. In line 220, the paper states 'the system’s invariant statistics, which is known to be difficult to characterize', any references as foundmentals sources?
2. Regarding the computational cost for high dimensional output space, the diagonal $Q$ matrix is applied. How does this diagonal matrix affect the performance, such as accuracy and stability?
3. Following question 2, how to optimize $L$ as a learnable component to construct positive definite  $Q$? This is a strong prior, also affecting the optimization landscape for other model components. Can it be optimized accordingly with the overall stochastic gradient descent on the model?
4. As energy level $c$ is an important component in the model, the  paper explains fixing $c \gg 1$ . But in Figure 3, the choice of c seems to be system-dependent. If so, what is the purpose of fixing $c$, and how does it generalize to different chaos systems? If not, how sensitive is the model to the choice of $c$?
5. How does the paper derive the PCA of higher-dimensional chaotic systems like the NS systems in Figure 5, of which 2D spatial information analysis is more complicated?

Happy to consider increasing my score if the paper is improved to address all the questions and weakness concerns.

---

> ### Author Response · Authors · 2025-11-13
>
> > The paper has the following strengths worth credits: 1. … 2. … 3. …
>
> Thank you for acknowledging our key contributions in many aspects. We are not sure why this is not reflected in the reviewer’s rating.
>
> > The novelty of ellipsoidal constraints is incremental compared with Markov neural operator (MNO). ECO replaces the spherical hard constraint with ellipsoidal constraints and introduces a differentiable projection layer.
>
> The reviewer’s statement about novelty is unfair and factually incorrect for two reasons:
> (1) Our method self-discovers the constraint from data during training and guides the predicted system according to energy dissipation, while MNO hard-coded the spherical constraint based on the prior information about the true system. Removing the requirement for this critical information is an important contribution.
> (2) Our approach provides theoretical guarantees by leveraging control theory designs, compared to MNO’s spherical constraint which implements an artificial constant scaling of the current state and lacks formal guarantees.
>
> > Experiments and evaluations are limited to compare ECO with DeepONet…assess the effectiveness of ECO compared with existing constrained methods for chaotic systems.
>
> > Though the paper provides several metrics … Including these metrics would enhance the comprehensiveness of the evaluation.
>
> The core contribution of our work is a framework that establishes formal guarantees based on an unconstrained neural operator backbone model. Additional benchmarking against existing methods or additional statistical metrics would not affect our main message.
>
> > The Ablation study is insufficient…
>
> While we can add an ablation study on hyperparameter tuning, it would not affect our core contributions. In addition, the suggested ablation without the energy function would not make sense because the projection layer is built on the energy function.
>
> > Question 1
>
> See Stuart & Humphries, 1998
>
> > Question 2 & 3
>
> Empirically, we found that using the diagonal matrix helps with training stability.
>
> The construction of a positive definite $Q$ follows standard Cholesky decomposition, where the diagonal elements of the lower-triangular matrix $L$ are parameterized as exponentials of learnable parameters.
>
> > Question 4
>
> Fixing c improves training stability by removing redundant local optimality points where $Q$ and $c$ can scale together. In practice, it affects the size of the learned invariant set.
>
> > Question 5
>
> We consider the 2D domain as a concatenated vector and perform PCA analysis, similar to other results in the literature.

---

### Note · Authors · 2025-11-13

**Comment:**

We are withdrawing the paper from this venue. For the public record, we have responded to the reviewers’ comments below.

**Withdrawal Confirmation:**

I have read and agree with the venue's withdrawal policy on behalf of myself and my co-authors.